# Maternal Serum Inhibin-A Augments the Value of Maternal Serum PlGF and of sFlt-1/PlGF Ratio in the Prediction of Preeclampsia and/or FGR Near Delivery— A Secondary Analysis

Adi Sharabi-Nov [1,†], Tanja Premru Sršen [2,3,†], Kristina Kumer [4,5,†], Vesna Fabjan Vodušek [2], Teja Fabjan [4,5], Nataša Tul [3,6], Hamutal Meiri [7,*], Kypros H. Nicolaides [8,†] and Joško Osredkar [4,5,†]

[1] Ziv Medical Center, Safed, and Tel Hai College, Tel Hai 13100, Israel; adi_nov@hotmail.com
[2] Department of Perinatology, Division of Obstetrics and Gynecology, University Medical Center, Zaloška cesta 2, 1000 Ljubljana, Slovenia; tanja.premru@kclj.si (T.P.S.); Vesna.fabjan@mf.uni-lj.si (V.F.V.)
[3] Faculty of Medicine, University of Ljubljana, Vrazov trg 2, 1000 Ljubljana, Slovenia; natasa.tul@guest.arnes.si
[4] Institute of Clinical Chemistry and Biochemistry, University Medical Centre, Njegoševa 4, 1000 Ljubljana, Slovenia; kristina.kumer@kclj.si (K.K.); fabjan.teja@kclj.si (T.F.); josko.osredkar@kclj.si (J.O.)
[5] Faculty of Pharmacy, University of Ljubljana, cesta 7, 1000 Ljubljana, Slovenia
[6] Women's Hospital, Prečna ulica 4, 6230 Postojna, Slovenia
[7] TeleMarpe Ltd., 41 Beit El St., Tel Aviv 6908742, Israel
[8] The Fetal Medicine Research Institute, King's College Hospital, 16-20 Windsor Walk, London SE5 8BB, UK; kypros@fetalmedicine.com
* Correspondence: hamutal62@hotmail.com; Tel.: +972-54-7774762
† These authors contributed equally to this work.

**Abstract:** Objective: We previously provided evidence to confirm that maternal serum levels of soluble Fms-like tyrosine kinase-1 (sFlt-1), placental growth factor (PlGF), and their ratio are useful tools to direct the management of preeclampsia (PE), fetal growth restriction (FGR), and PE+FGR near delivery. In this secondary analysis, we further examine the potential additive value of maternal serum Inhibin-A, which is a hormone marker of the transforming growth factor family, to the accuracy provided by maternal serum PlGF and sFlt-1. Methods: We conducted a secondary analysis where we extracted the data of a cohort of 125 pregnant women enrolled near delivery at the clinics of the University Medical Center of Ljubljana, Slovenia. The dataset included 31 cases of PE, 16 of FGR, 42 of PE+FGR, 15 preterm delivery (PTD), and 21 unaffected controls with delivery of a healthy baby at term. Cases delivered before 34 weeks' gestation included 10 of PE, 12 of FGR, 28 of PE+FGR, and 6 of PTD. In addition to the recorded demographic characteristics and medical history and the maternal serum levels of PlGF and sFlt-1/PlGF ratio, which were previously published, we evaluated the added value of maternal serum Inhibin-A. The predictive accuracy of each biomarker, their ratios, and combinations were estimated from areas under the curve (AUC) of receiver operating characteristics (ROC) curves, Box and Whisker plots, and by multiple regression. We estimated accuracy by the continuous marker model and a cutoff model. Results: In this study, we combined Inhibin-A with PlGF or with the sFlt-1/PlGF ratio and showed a 10–20% increase in AUCs and 15–45% increase in the detection rate, at 10% false positive rate, of PE, and a lower, but significant, increase for PE+FGR and FGR in all cases but not for FGR in early cases delivered < 34 weeks. The use of a cutoff model was adequate, although a bit higher accuracy was obtained from the continuous model. The highest correlation was found for PlGF with all three complications. Conclusion: In this secondary analysis, we have found that maternal serum Inhibin-A improves the accuracy of predicting PE and PE+FGR provided by maternal serum angiogenic markers alone, bringing the results to a diagnostic level; thus, it could be considered for directing clinical management. Inhibin-A had smaller or no added value for the accuracy of predicting FGR alone, mainly of early cases delivered <34 weeks.

**Keywords:** preeclampsia; fetal growth restriction; sFlt-1; Inhibin-A; placental growth factor

## 1. Introduction

Preeclampsia (PE) is a major pregnancy disorder unique to pregnancy that is associated with new-onset hypertension, which occurs most often after 20 weeks of gestation and frequently near term. Although often accompanied by new-onset proteinuria, hypertension and other signs or symptoms of preeclampsia may present in some women in the absence of proteinuria [1–10]. The condition affects 2–7% of pregnancies, and worldwide, it is accompanied by one maternal death every 8 min and a yearly loss of 500,000 fetuses [6–10]. Preeclampsia presents either alone or in combination with fetal growth restriction (PE+FGR) [1]. Fetal growth restriction (FGR) is a common pregnancy complication where the fetus does not grow to its expected biological potential in utero. It develops when the placenta fails to deliver an adequate supply of oxygen and nutrients to the developing fetus, which is termed placental insufficiency, and/or as a result of individual's genetic makeup, nutrient availability from the mother, and environmental factors, coupled with the capacity of the placenta to adequately transfer nutrients and oxygen to the fetus. Endocrine modulation of these interactions is the basic determinant of fetal growth [11–13]. The successful management of PE and/or FGR improves pregnancy outcome and reduces life-long complications [1,2,9–13]. Both PE and FGR can result in preterm delivery (PTD); there are many similarities between early onset PE and/or FGR and PTD itself, because all three often require emergency delivery by cesarean section, and they are associated with low birth weight and neonatal complications due to prematurity [14,15].

Several biochemical markers emerged as being useful in the clinical management of women admitted to hospital with suspected PE and/or FGR, including a reduced maternal serum level of the proangiogenic placental growth factor (PlGF), which is a hormone reflecting placental size and increased level of the anti-angiogenic soluble Fms-like tyrosine kinase-1 (sFlt-1), the soluble form of the receptor to epidermal growth factor or increased sFlt-1/PlGF ratio [16–22]. Similar results were also found in our dataset [23–25].

This study is a secondary analysis of a previously published dataset. Here we evaluated whether adding maternal serum level of Inhibin-A, a glycoprotein hormone belonging to the transforming growth factor family [26,27], could elevate the prediction accuracy of the complications of PE, FGR, and PE+FGR by combined analysis with PlGF and sFlt-1/PlGF ratio. Inhibin-A is abundantly expressed in the placenta, and as we [24] and others [26,27] have previously reported, in cases of PE and/or FGR, the level of Inhibin-A is significantly elevated in the placenta, in the uterine vein collecting biomolecules released from the placenta, and in the maternal circulation. Our emphasis in this study was to further extract the medical records and the maternal serum levels of the biomarkers to explore if there is a potential added value of combining Inhibin-A with PlGF and/or with sFlt-1/PlGF ratio for the accurate prediction of suspected PE and/or FGR.

## 2. Sample and Methods

### 2.1. Sample

We performed a secondary analysis from a dataset of patients who were enrolled between 2012 and 2015 after obtaining approval of the National Medical Ethics Committee of the Republic of Slovenia (Approval on 4 December 2011, by approval No. 104/04/12). Recruitment after signing on the informed consent was made at the outpatient clinics of high-risk pregnancies at the Department of Perinatology of the University Medical Centre of Ljubljana, Slovenia. All patients were not in labor when included in the study, and their gestational age was 24 weeks or more. The cohort included patients 18 years old and above with singleton viable pregnancy without major fetal anomalies, or pre-existing renal, hematological, or autoimmune conditions. Gestational age was determined from ultrasound measurements of the fetal crown–rump length in the first trimester of pregnancy [28]. Clinical management adhered to hospital guidelines. All patients were Caucasian.

The study population included 31 cases of PE, 16 of FGR, 42 PE+FGR cases, 15 of PTD, and 21 unaffected cases who delivered a healthy baby at term. The cases that were

delivered at <34 weeks included 10 of PE, 12 of FGR, 22 of PE+FGR, and 6 of PTD in the absence of PE and/or FGR or placental abruption as was previously described [23–25].

### 2.2. Biochemical and Biophysical Markers

All biomarkers were tested from serum samples collected at the time of enrolment and measured at the chemical pathology laboratory of the university. Serum PlGF and sFlt-1 were measured by the Elecsys analyzer (Cobas e411 system, Roche Diagnostics, Roche Diagnostics, Mannheim, Germany) according to the manufacturer instructions [17–19]. Inhibin-A was measured by the Access 2 immunoassay analyzer (Access 2 Immunoassay System (1049), Beckman Coulter, Brea, CA, USA) according to the manufacturer's instructions [24].

Blood pressure was measured according to the guidelines of the Fetal Medicine Foundation using a calibrated electronic device, and the mean arterial blood pressure (MAP) was calculated as (systolic + diastolic × 2)/3 [29].

### 2.3. Outcome Measures

In this paper, we used the updated criteria for the definition of preeclampsia as published in June 2020 by the American College of Obstetrics and Gynecology (ACOG) and of the International Society for the Study of Hypertension Disorders of Pregnancy (ISSHP) [1,2]. Preeclampsia (PE) was defined as systolic blood pressure of 140 mm Hg or more or diastolic blood pressure of 90 mm Hg or more on two occasions at least 4 h apart after 20 weeks of gestation in a woman with a previously normal blood pressure. (Severe PE was defined according to systolic blood pressure of 160 mm Hg or more or diastolic blood pressure of 110 mm Hg or more. For the latter, the blood pressure was confirmed within a shorter interval (minutes) to facilitate timely management) [29]. The new onset proteinuria was defined as 300 mg or more per 24 h urine collection (or this amount extrapolated from a timed collection) or protein/creatinine ratio of 0.3 mg/dL or more or dipstick reading of 2+ (used only if other quantitative methods not available) [30]. In the absence of proteinuria, new-onset hypertension with the new onset of any of the following: thrombocytopenia defined as platelet count less than $100 \times 10^9$ /L [31], renal insufficiency was determined as serum creatinine concentrations greater than 1.1 mg/dL or a doubling of the serum creatinine concentration in the absence of other renal disease, and impaired liver function was defined as elevated blood concentrations of liver transaminases to twice normal concentration [32,33]. Other symptoms included pulmonary edema, new-onset headache unresponsive to medication, and those not accounted for by alternative diagnoses or visual symptoms [1,2]. Given the new ACOG and ISSHP definition of preeclampsia published after the study was completed [1,2], we reviewed the database on a patient-by-patient basis to verify that patients included in the preeclampsia group according to our hospital clinical guidelines comply with these new ACOG and ISSHP definitions. Luckily, we were able to reassure no changes to patients' clinical definitions while adopting the updated criteria.

Fetal growth restriction was defined according to the definition of the International Society of Ultrasound in Obstetrics and Gynecology (ISUOG) as sonographic estimated fetal weight below the 10th percentile [34,35], and abnormal blood flow patterns demonstrated by Doppler ultrasound in the uterine, umbilical, or middle cerebral arteries [11–13,36].

Preterm delivery was defined as delivery <37 weeks' gestation after the spontaneous onset of labor or spontaneous preterm pre labor rupture of membranes (PPROM) but not due to any of PE, FGR or PE+FGR, fetal abnormalities, or chorioamnionitis [14,15].

### 2.4. Statistical Analyses

The median with 95% Confidence Interval (95% CI) was calculated for each marker, and each adverse pregnancy outcome group was compared to results from the normal term delivery group using Mann–Whitney test. Kruskal–Wallis analysis was performed to calculate the difference among multiple study groups. Both Mann Whitney and Kruskal–

Wallis analysis were performed with the SPSS package version 24 (SPSS Inc., Chicago, IL, USA). Bonferroni post hoc corrections for multiple comparisons were entered. Box-Plot graphs provided the graphic description of medians and quartile distribution. Receiver operating characteristic (ROC) curves were used to calculate the area under the curve (AUC) from marker values or from their ratios with 95% CI and to calculate the detection rate at 10% fixed false positive rate (FPR). The positive predictive value (PPV) was calculated as true cases at the cutoff divided by all cases at the cutoff, and the negative predictive value (NPV) was calculated as all true negative cases at the cutoff divided by all negative cases at the cutoff. In the continuous model, the AUCs were extracted from the ROC curves. In the cutoff model, AUC and detection rate were extracted from cutoffs. Combined analysis was performed by combining percentiles of individual marker values for each FPR. Where possible, we used curve fitting by polynomial calculation to smooth ROC curves.

## 3. Results

### 3.1. Cohort Characteristics

This secondary analysis included 31 cases of PE, 16 of FGR, 42 of PE+FGR, 15 preterm delivery (PTD), and 21 unaffected controls with delivery of a healthy baby at term. Cases delivered before 34 weeks' gestation included 10 of PE, 12 of FGR, 28 of PE+FGR, and 6 of PTD. Data were collected from all so there were no loss to follow up or those who dropped consent.

Cohort features were previously described [24]. Groups had similar maternal age and parity. Gestational ages at enrollment and when all marker testing was performed for reporting in this manuscript were 34 weeks for the group of term delivery, 31 weeks for the PTD group delivered <37 weeks, 31.9 weeks for PE, 31.4 weeks for FGR, and 31.8 weeks for PE+FGR. The groups of PE and PE+FGR had higher body mass index (BMI); in the PE and FGR groups, there was a higher incidence of conception by in vitro fertilization (IVF) and in the PE group, there was a higher incidence of patients with history of previous PE, diabetes mellitus, or polycystic ovary syndrome. The blood pressure at presentation was 150/94 in the PE group, 151/94 in the PE+FGR group, 131/80 in the FGR group, 119/76 in the PTD group, and 112/71 in the term delivery controls. Gestational age at delivery was 34.2, 31.7, 32.0, and 33.8 weeks in the PE, FGR, PE+FGR, and PTD groups, respectively, compared to 39.1 for term delivery control group. Baby birthweights were 2306, 1306, 1449, and 2207 g in the PE, FGR, PE+FGR, and PTD groups, respectively, compared to 3300 g for the term delivery control group.

### 3.2. Median Marker Levels in the Outcome Groups

In all the cases in the PE, FGR, and PE+FGR groups, compared to the unaffected term delivery controls and PTD <37 weeks, the median maternal serum levels of Inhibin-A, sFlt-1/PlGF ratio, and Inhibin-A/PlGF ratio were significantly higher, and PlGF was significantly lower (Figure 1 and Table 1). There was good separation between affected and unaffected pregnancies at a cutoff of 1000 pg/mL for Inhibin-A, 200 pg/mL for PlGF, 38 for the sFlt-1/PlGF ratio, and 7 for the Inhibin-A/PlGF ratio. The cutoffs place >90% of the unaffected and of the PTD group (corresponding to 10% FPR) in one side and >80% of the patients with complications on the other side of the line. Similarly, in the cases of PE, FGR, and PE+FGR delivered < 34 weeks, compared to the group with PTD < 34 weeks, the median Inhibin-A, sFlt-1/PlGF ratio, and Inhibin-A/PlGF ratio were significantly higher and PlGF was lower.

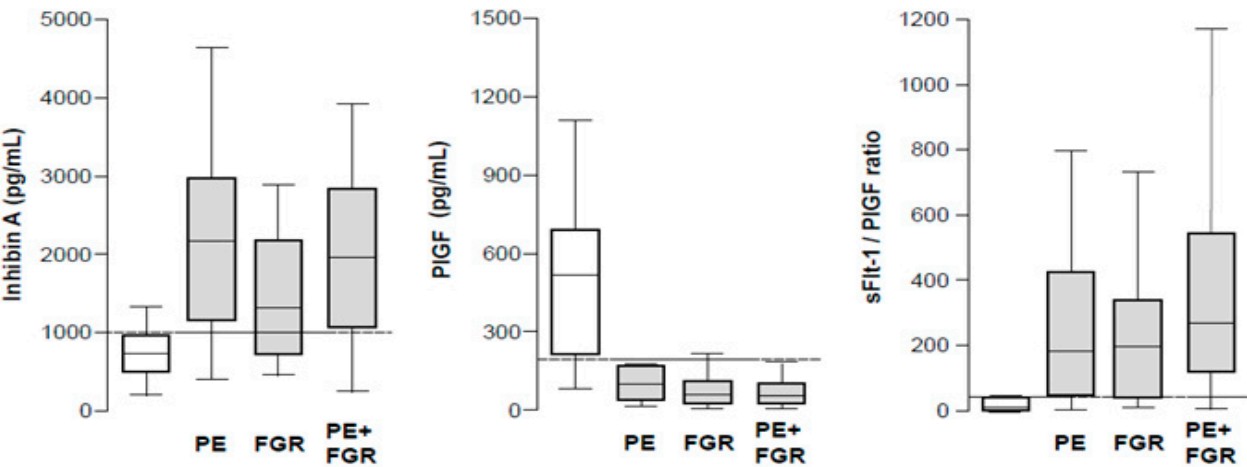

**Figure 1.** Comparison of Biomarker levels. Box and whiskers' plots of Inhibin-A, placental growth factor (PlGF) and soluble Fms-like protein kinase-1 (sFlt-1)/PlGF ratio in all cases of PE (*n* = 31), FGR (*n* = 16) and PE+FGR (*n* = 42). The horizontal lines indicate the cutoffs that separate the affected cases (gray histograms) from unaffected controls (white histograms) that were 90% to 10% divided between the cutoff-line.

**Table 1.** Pregnancy and Biomarkers Characteristics.

| Characteristic | Term Delivery | Birth < 37 Weeks | Preeclampsia (PE) | Fetal Growth Restriction (FGR) | PE+FGR | *p*-Value |
|---|---|---|---|---|---|---|
| **All Participants** | | | | | | |
| Number of Patients | 21 | 15 | 31 | 16 | 42 | |
| Gestational Age at Enrollment (weeks) | 34.0 [32.0–35.9] | 31.2 [29.4–32.9] * | 33.9 [32.3–35.6] | 31.4 [29.1–33.6] * | 31.8 [30.7–32.8] * | 0.027 |
| Maternal Age (years) | 31.6 [29.5–33.8] | 31.3 [29.7–32.9] | 32.0 [29.9–34.1] | 31.7 [29.7–33.7] | 32.9 [31.1–34.7] | 0.792 |
| Body Mass Index (kg/m$^2$) | 25.8 [23.7–27.9] | 24.6 [22.9–26.4] | 29.5 [26.5–32.6] * | 27.6 [24.2–31.0] | 29.6 [26.9–32.4] * | 0.011 |
| Previous PE (%) | 4.8 | 6.7 | 6.5 | 6.3 | 9.5 | 0.965 |
| Chronic Hypertension (%) | 0 | 0 | 19.4 * | 0 | 16.7 * | 0.032 |
| Diabetes (%) | 0 | 0 | 3.2 | 0 | 4.8 | 0.787 |
| Polycystic Ovary (%) | 0 | 0 | 0 | 0 | 7.1 * | 0.204 |
| Parity | 1.7 [1.3–2.0] | 1.6 [1.2–2.1] | 1.4 [1.0–1.8] | 1.5 [1.1–1.9] | 1.5 [1.2–1.8] | 0.806 |
| Conception by IVF (%) | 4.8 | 0 | 6.5 | 0 | 11.9 * | 0.361 |
| MAP (mm HG) | 85 [80–90] | 90 [82–98] | 113 [109–116] ** | 97 [93–101] * | 113 [109–116] ** | <0.001 |
| UTPI | 0.68 [0.66–0.70] | 0.70 [0.64–0.61] | 0.80 [0.60–1.17] * | 1.35 [1.05–1.66] ** | 1.42 [1.25–1.56] ** | <0.001 |
| Inhibin-A (pg/mL) | 724 [491–904] | 330 [261–928] | 2097 [1546–2660] * | 1269 [760–2348] * | 1876 [1239–2295] * | <0.001 |
| sFlt-1/PlGF | 5 [3–31] | 6 [2–9] | 177 [106–301] * | 195 [55–310] * | 265 [168–382] * | <0.001 |
| PlGF (pg/mL) | 524 [223–681] | 693 [308–980] | 101 [69–153] * | 76 [43–117] * | 62 [48–87] * | <0.001 |
| Inhibin-A/PlGF | 3.1 [0.7–3.6] | 1.2 [0.2–1.2] | 41.0 [10.2–39.4] ** | 36.3 [11.2–50.8] ** | 45.0 [19.5–44.1] ** | 0.001 |
| Gestational Age at Delivery (weeks) | 39.1 [38.5–39.7] | 33.8 [32.1–35.5] * | 34.2 [32.6–35.9] * | 31.7 [29.4–34.0] ** | 32.0 [31.0–33.1] ** | <0.001 |
| Delivery by C-Section (%) | 23.8 | 30.8 | 54.8 ** | 60.0 * | 83.4 ** | <0.001 |
| Baby's Birthweight (grams) | 3330 [3133–3528] | 2207 [1872–2542] * | 2306 [1906–2705] * | 1306 [834–1778] ** | 1449 [1247–1651] ** | <0.001 |

**Table 1.** *Cont.*

| Characteristic | Term Delivery | Birth < 37 Weeks | Preeclampsia (PE) | Fetal Growth Restriction (FGR) | PE+FGR | *p*-Value |
|---|---|---|---|---|---|---|
| | | | **Birth < 34 weeks** | | | |
| Number of Patients | | 6 | 10 | 12 | 28 | |
| Gestationa Age at Enrollment (wks) | | 29.2 [26.8–31.6] | 29.9 [27.5–32.3] | 29.3 [27.7–30.8] | 29.9 [28.9–30.9] | 0.805 |
| Maternal Age (years) | | 31.3 [27.8–34.8] | 33.8 [33.0–37.7] | 31.5 [29.2–33.8] | 33.1 [30.7–35.5] | 0.668 |
| Body Mass Iindex (kg/meter2) | | 24.7 [21.0–28.4] | 30.7 [26.2–35.2] | 26.3 [23.9–28.8] | 29.7 [26.1–33.4] | 0.123 |
| Previous PE (%) | | 0 | 0 | 0 | 7.1 | 0.591 |
| Chronic Hypertension (%) | | 0 | 7.7 | 0 | 21.4 * | 0.146 |
| Diabetes (%) | | 0 | 0 | 0 | 3.6 | 0.771 |
| Polycystic Ovary (%) | | 0 | 0 | 0 | 3.6 | 0.771 |
| Parity | | 1.8 [1.0–2.6] | 1.5 [0.7–2.4] | 1.3 [0.9–1.7] | 1.6 [1.2–2.0] | 0.807 |
| Conception by IVF (%) | | 0 | 15.4 * | 0 | 10.7 | 0.498 |
| MAP (mmHg) | | 87 [70–103] | 114 [107–121] | 96 [91–101] | 115 [110–119] * | >0.001 |
| UTPI | | 0.69 [0.57–0.80] | 1.20 [0.83–1.57] * | 1.62 [1.35–1.90] ** | 1.43 [1.27–1.58] ** | >0.001 |
| Inhibin-A (pg/mL) | | 457 [0–1015] | 3216 [2212–4220] ** | 1503 [1019–1987] * | 238 [1711–3057] ** | 0.003 |
| sFlt-1/PlGF | | 6 [0–13] | 521 [246–796] * | 307 [174–439] * | 460 [273–647] * | 0.050 |
| PlGF (pg/mL) | | 762 [182–1343] | 215 [0–479] * | 70 [27–113] ** | 103 [39–167] ** | >0.001 |
| Inhibin-A/PlGF | | 1.1 [0.2–3.5] | 75.0 [17.1–114.2] ** | 44.8 [11.2–70.3] * | 57.7 [19.8–66.0] ** | 0.049 |
| GA at delivery (wks) | | 31.0 [28.0–34.0] | 30.2 [27.8–32.6] | 29.5 [28.0–31.1] | 30.2 [29.2–31.2] | 0.805 |
| Delivery by C-Section (%) | | 20.0 | 77.9 * | 72.7 * | 92.6 ** | 0.003 |
| Baby Birthweight (grams) | | 1669 [1318–2020] | 1276 [923–1628] * | 874 [627–1121] ** | 1171 [995–1346] * | 0.018 |

The Medians (95% Confidence Interval (CI)) of pregnancy characteristics and maternal serum levels of the biochemical markers in the different patients' groups classified according to pregnancy outcome. The *p* values (column to the right) among all groups were calculated with Kruskal–Wallis non-parametric test and corrected by Bonferroni post hoc corrections for multiple comparisons. In addition, each complication was compared by Mann–Whitney non-parametric test to the results of term delivery in the upper part and to birth <34 weeks in the lower part, and the asterisk on the right side of the number represents * $p < 0.05$, ** $p < 0.01$ calculated by this analysis. sFlt-1—soluble Fms-like tyrosine kinase-1, PlGF—placental growth factor. IVF—conception by in vitro fertilization. PE—preeclampsia, FGR—fetal growth restriction, MAP—mean arterial blood pressure, UTPI—uterine artery pulsatility index by Doppler sonography.

### 3.3. AUC Analysis

The AUCs and detection rates at 10% FPR for PE, FGR, and PE+FGR are shown in Table 2. Table 3 and Figures 2 and 3 demonstrate that a combination of PlGF and Inhibin-A was superior to PlGF alone in the prediction of all PE, all FGR, all PE+FGR, and PE < 34 weeks, but not FGR or PE+FGR < 34 weeks. Similarly, a combination of sFlt-1/PlGF ratio plus Inhibin-A was superior to sFlt-1/PlGF ratio alone in the prediction of all PE, all FGR, all PE+FGR, PE < 34 weeks, and PE+FGR < 34 weeks, but not FGR < 34 weeks.

**Table 2.** Accuracy prediction of pregnancy complications by the Continuous and Cutoff model.

| Condition | Marker | Continuous Model | | | Cutoff Model | | | |
|---|---|---|---|---|---|---|---|---|
| | | AUC (95% CI) | DR at 10% FPR | Cutoff | AUC (95% CI) | DR at 10% FPR | PPV | NPV |
| All PE (n = 31) | Inhibin-A | 0.91 (0.84–0.98) | 72 | 1000 pg/mL | 0.80 (0.69–0.92) | 42 | 79 | 81 |
| | PlGF | 0.85 (0.75–0.95) | 53 | 200 pg/mL | 0.82 (0.71–0.93) | 43 | 82 | 83 |
| | sFlt-1/PlGF | 0.89 (0.80–0.97) | 79 | 38 | 0.85 (0.74–0.96) | 68 | 85 | 83 |
| | Inhibin-A/PlGF | 0.92 (0.85–0.99) | 79 | 7 | 0.83 (0.72–0.94) | 73 | 91 | 79 |
| All FGR (n = 16) | Inhibin-A | 0.82 (0.70–0.95) | 50 | 1000 pg/mL | 0.75 (0.59–0.91) | 36 | 65 | 84 |
| | PlGF | 0.95 (0.89–1.00) | 77 | 200 pg/mL | 0.86 (0.75–0.98) | 68 | 74 | 94 |
| | sFlt-1/PlGF | 0.97 (0.92–1.00) | 81 | 38 | 0.86 (0.73–0.99) | 69 | 76 | 91 |
| | Inhibin-A/PlGF | 0.94 (0.88–1.00) | 75 | 7 | 0.84 (0.71–0.98) | 76 | 86 | 88 |
| All PE+FGR (n = 42) | Inhibin-A | 0.87 (0.78–0.95) | 68 | 1000 pg/mL | 0.80 (0.69–0.90) | 41 | 84 | 74 |
| | PlGF | 0.92 (0.86–0.98) | 71 | 200 pg/mL | 0.87 (0.78–0.96) | 71 | 88 | 85 |
| | sFlt-1/PlGF | 0.97 (0.93–1.00) | 93 | 38 | 0.92 (0.84–0.99) | 80 | 90 | 91 |
| | Inhibin-A/PlGF | 0.94 (0.88–1.00) | 85 | 7 | 0.90 (0.82–0.98) | 86 | 95 | 83 |
| PE < 34 w (n = 10) | Inhibin-A | 0.98 (0.93–1.00) | 91 | 400 pg/mL | 0.90 (0.68–1.00) | 49 | 92 | 100 |
| | PlGF | 0.89 (0.73–1.00) | 60 | 300 pg/mL | 0.91 (0.76–1.00) | 53 | 100 | 71 |
| | sFlt-1/PlGF | 0.93 (0.80–1.00) | 82 | 120 | 0.91 (0.76–1.00) | 82 | 100 | 71 |
| | Inhibin-A/PlGF | 0.96 (0.88–1.00) | 91 | 2 | 0.86 (0.62–1.00) | 45 | 100 | 71 |
| FGR < 34 w (n = 12) | Inhibin-A | 0.90 (0.71–1.00) | 50 | 400 pg/mL | 0.90 (0.68–1.00) | 50 | 92 | 100 |
| | PlGF | 1.00 (1.00–1.00) | 100 | 300 pg/mL | 1.00 (1.00–1.00) | 100 | 100 | 100 |
| | sFlt-1/PlGF | 1.00 (1.00–1.00) | 100 | 120 | 0.92 (0.78–1.00) | 85 | 100 | 71 |
| | Inhibin-A/PlGF | 0.98 (0.93–1.00) | 92 | 2 | 0.90 (0.68–1.00) | 50 | 100 | 71 |
| PE+FGR < 34 w (n = 28) | Inhibin-A | 0.93 (0.80–1.00) | 67 | 400 pg/mL | 0.88 (0.67–1.00) | 100 | 100 | 80 |
| | PlGF | 0.96 (0.90–1.00) | 100 | 300 pg/mL | 0.96 (0.90–1.00) | 100 | 100 | 71 |
| | sFlt-1/PlGF | 1.00 (1.00–1.00) | 100 | 120 | 0.91 (0.81–1.00) | 83 | 100 | 50 |
| | Inhibin-A/PlGF | 0.99 (0.97–1.00) | 96 | 2 | 0.88 (0.67–1.00) | 47 | 100 | 63 |

Prediction of all cases of preeclampsia (PE), fetal growth restriction (FGR), and PE+FGR and of PE, FGR and PE+FGR delivered < 34 weeks, by maternal serum levels of Inhibin-A, placental growth factor (PlGF) soluble Fms-like tyrosine kinase 1 (sFlt-1)/PlGF ratio and Inhibin-A/PlGF ratio. DR—detection rate, FPR—false positive rate, AUC—area under the curve, PPV and NPV—positive and negative predictive values.

**Table 3.** Combined Marker Analysis.

| Condition | Marker | AUC (95% CI) | *p* | DR at 10% FPR |
|---|---|---|---|---|
| All PE (*n* = 31) | PlGF | 0.85 (0.75–0.95) | <0.001 | 53 |
| | PlGF + Inhibin-A | 0.98 (0.90–1.00) | 0.006 | 94 |
| | sFlt-1/PlGF | 0.89 (0.80–0.97) | <0.001 | 79 |
| | sFlt-1/PlGF + Inhibin-A | 0.95 (0.91–0.99) | 0.003 | 87 |
| All FGR (*n* = 16) | PlGF | 0.95 (0.89–1.00) | <0.001 | 77 |
| | PlGF + Inhibin-A | 0.98 (0.87–1.00) | 0.002 | 93 |
| | sFlt-1/PlGF | 0.97 (0.92–1.00) | <0.001 | 81 |
| | sFlt-1/PlGF + Inhibin-A | 0.99 (0.94–1.00) | <0.001 | 90 |
| All PE+FGR (*n* = 42) | PlGF | 0.92 (0.86–0.98) | <0.001 | 71 |
| | PlGF + Inhibin-A | 0.98 (0.91–0.99) | <0.001 | 90 |
| | sFlt-1/PlGF | 0.97 (0.93–1.00) | <0.001 | 93 |
| | sFlt-1/PlGF + Inhibin-A | 0.99 (0.93–1.00 | 0.004 | 95 |
| PE < 34 w (*n* = 10) | PlGF | 0.89 (0.73–1.00) | 0.015 | 60 |
| | PlGF + Inhibin-A | 0.99 (0.91–1.00) | <0.001 | 98 |
| | sFlt-1/PlGF | 0.93 (0.80–1.00) | 0.008 | 82 |
| | sFlt-1/PlGF +Inhibin-A | 0.99 (0.89–1.00) | 0.009 | 98 |
| FGR < 34 w (*n* = 12) | PlGF | 1.00 (1.00–1.00) | <0.001 | 100 |
| | PlGF + Inhibin-A | No added value | | No added value |
| | sFlt-1/PlGF | 1.00 (1.00–1.00) | <0.001 | 100 |
| | sFlt-1/PlGF + Inhibin-A | No added value | | No added value |
| PE+FGR < 34 w (*n* = 28) | PlGF | 0.96 (0.90–1.00) | <0.001 | 100 |
| | PlGF + Inhibin-A | No added value | | No added value |
| | sFlt-1/PlGF | 1.00 (1.00–1.00) | <0.001 | 100 |
| | sFlt-1/PlGF + Inhibin-A | No added value | | No added value |

Comparison of screening accuracy of combined analysis by placental growth factor (PlGF) plus Inhibin-A versus PlGF alone and between soluble Fms-like tyrosine kinase-1 (sFlt-1)/ratio plus Inhibin-A versus sFlt-1/PlGF ratio alone. DR—detection rate, FPR—false positive rate, AUC—area under the curve, *p*—statistical significance against the arbitrary line of AUC = 0.5. PE—preeclampsia, FGR—fetal growth restriction.

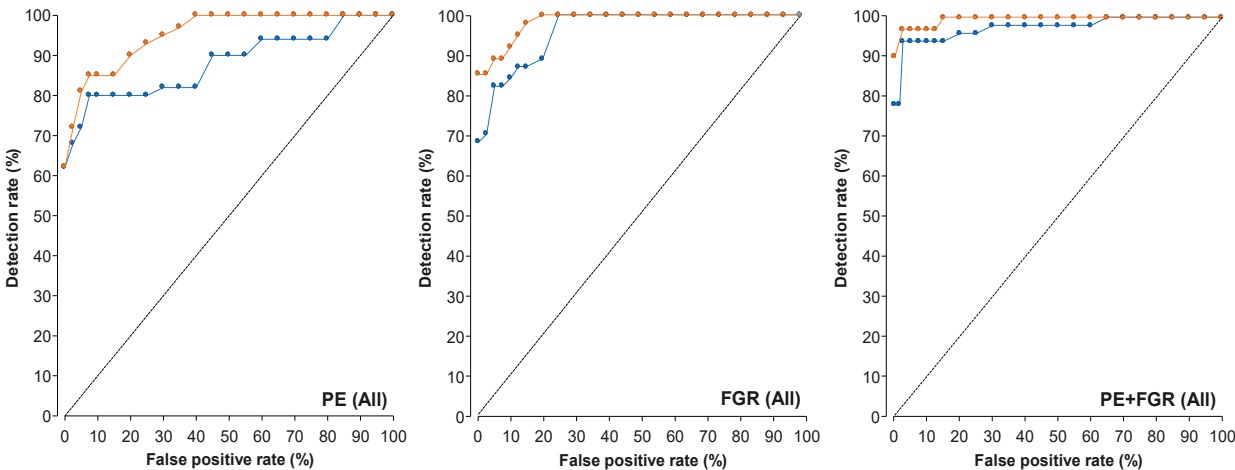

**Figure 2.** Combined Analysis of markers pairs for all participants in each group (sflt-1/PlGF + Inhibin-A). Single marker analysis by the ratio of soluble Fms-like tyrosine kinase-1 to placental growth factor (PlGF) (sFlt-1/PlGF) depicted in the blue lines and dotes, and the pair markers analysis made by combining sFlt-1/PlGF with Inhibin -A (orange lines and dotes). Left—preeclampsia (PE), middle—fetal growth restriction (FGR), Right—PE+FGR. The arbitrary line (black) corresponds to AUC = 0.5.

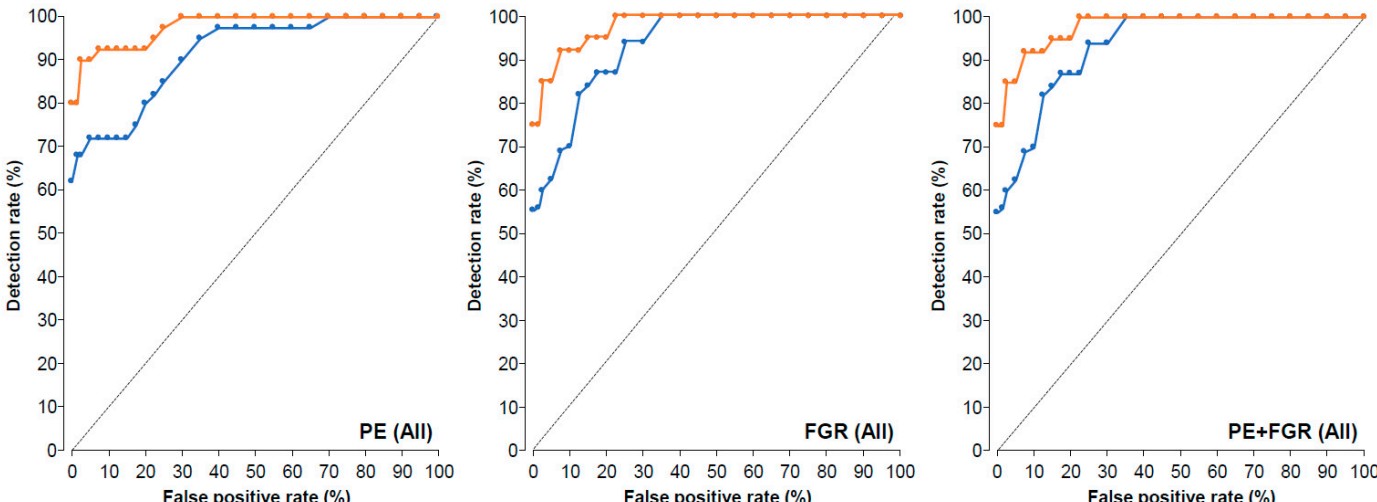

**Figure 3.** Combined Analysis of markers pairs for all participants in each group (PlGF + Inhibin-A). Single marker analysis placental growth factor (PlGF) depicted in the blue lines and dotes, and the pair markers analysis made by combining PlGF with Inhibin -A (orange lines and dotes). Left—preeclampsia (PE), middle—fetal growth restriction (FGR), Right—PE+FGR. The arbitrary line (black) corresponds to AUC = 0.5.3.4. Multiple Regression.

Multiple regression analysis was performed with the SPSS statistical package used to assess whether gestational age (GA), birth weight (BW), mean arterial blood pressure (MAP), and PE, FGR, and PE+FGR can predict the marker level. The equations used were as follows.

$$\text{Inhibin-A} = -46.66 - 0.61 \times \text{BW} - 16.04 \times \text{MAP} + 104.06 \times \text{GA} + 1692 \times \text{PE} + 661 \times \text{FGR} + 1165.94 \times (\text{FGR} + \text{PE})$$

$$\text{PlGF} = 1101 + 0.17 \times \text{BW} + 0.79 \times \text{MAP} - 27.88 \times \text{GA} - 401 \times \text{PE} - 424 \times \text{FGR} - 381 \times (\text{FGR+PE})$$

$$\text{sFlt-1/PlGF} = 1297 - 0.05 \times \text{BW} - 6.08 \times \text{MAP} - 16.05 \times \text{GA} + 240 \times \text{PE} + 118 \times \text{FGR} + 210 \times (\text{FGR} + \text{PE})$$

The regression yielded statistical significance (regression coefficient $R^2 = 0.28$, F-Test parameters, and degree of freedom ($F_{(6,63)} = 3.63$, $p < 0.01$; $R^2 = 0.40$, $F_{(6,63)} = 6.40$, $p < 0.001$; $R^2 = 0.54$, $F_{(6,63)} = 11.22$, $p < 0.001$, for PE, FGR and PE+FGR, respectively). At all

three markers, the parameters of GA and BW were not significant predictors ($p > 0.05$). MAP was negatively and significantly associated only with sFlt-1/PlGF (standardized coefficients (β) = $-0.33$, $p < 0.01$) (Table 4).

**Table 4.** Multiple regression model to predict Inhibin-A, PlGF, and sFlt-1/PlGF ratio.

| | Inhibin-A | | | PlGF | | | sFlt-1/PlGF | | |
|---|---|---|---|---|---|---|---|---|---|
| **Variables** | **B** | **S.E.** | **β** | **B** | **S.E.** | **β** | **B** | **S.E.** | **β** |
| GA (weeks) | 104.06 | 88.83 | 0.34 | −27.88 | 24.77 | −0.29 | −16.05 | 12.57 | −1.28 |
| BW (g) | −0.61 | 0.39 | −0.51 | 0.17 | 0.11 | 0.47 | −0.05 | 0.06 | −0.24 |
| MAP (mm HG) | −16.04 | 14.16 | −0.16 | 0.79 | 3.86 | 0.26 | −6.08 | 1.96 | −0.33 ** |
| PE | 1692 | 493 | 0.49 ** | −401 | 135 | −0.37 ** | 240 | 69 | 0.39 ** |
| FGR | 661 | 568 | 0.19 | −424 | 157 | −0.40 ** | 118 | 80 | 0.19 |
| FGR+PE | 1166 | 506 | 0.39 * | −381 | 139 | −0.41 ** | 210 | 70 | 0.39 ** |
| $F_{(6,63)}$ | 3.63 ** | | | 6.40 *** | | | 11.22 *** | | |
| $R^2$ | 0.28 | | | 0.40 | | | 0.54 | | |

Multiple regression analysis to assess whether gestational age (GA), birth weight (BW), mean arterial blood pressure (MAP), preeclampsia (PE), fetal growth restriction (FGR), and PE+FGR could predict Inhibin-A, PlGF and sFlt-1/PlGF ratio where the complications are evaluated against the unaffected controls. B = unstandardized coefficients, S.E. = coefficients standard error, β = standardized coefficients, $R^2$ + Regression Coefficient, $F_{(6,63)}$ = F Test for six parameters and 63 degrees of freedom. Asterisks present: * $p < 0.05$, ** $p < 0.01$ and *** $p < 0.001$.

For Inhibin-A, there were positive significant correlations with PE and FGR+PE (β = 0.49, $p = 0.001$ and β = 0.39, $p < 0.05$, respectively). There were also positive correlations between sFlt-1/PlGF ratio and PE and FGR+PE (β = 0.39, $p < 0.01$ for both). For PlGF, there was a negative significant correlation with each of the three complications of PE, FGR, and PE+FGR ($p < 0.01$).

## 4. Discussion

### 4.1. Main Findings

The study has investigated the potential added value of maternal serum Inhibin-A to the accuracy of predicting PE, FGR, and PE+FGR by the best angiogenic markers: PlGF and sFlt-1/PlGF ratio near the time of delivery. We found that first, on its own, maternal serum Inhibin-A is a moderately good biomarker of PE and PE+FGR, but not of FGR alone; second, combining Inhibin-A with PlGF, compared to PlGF alone, was associated with a 13% and 41% improvement in the AUC and detection rate at 10% FPR of all PE with respective values of 10% and 37% for early PE; third, combining Inhibin-A with PlGF, compared to PlGF alone, was associated with a 6% and 29% improvement in the AUC and detection rate at 10% FPR of all PE+FGR, respectively, but there was no benefit in the prediction of early PE+FGR; fourth, the addition of Inhibin-A had low or no added value to PlGF in the prediction of FGR alone, and this is consistent with the finding of a high correlation between PlGF and all three complications, whereas Inhibin-A was correlated with PE and PE+FGR, but not FGR alone; fifth, combining Inhibin-A with the sFlt-1/PlGF ratio, compared to the sFlt-1/PlGF ratio alone, was associated with a 6% and 8% improvement in the AUC and detection rate at 10% FPR of all PE with respective 6% and 16% increase for early PE; sixth, combining Inhibin-A with the sFlt-1/PlGF ratio, compared to the sFlt-1/PlGF ratio alone, there was a minimal impact on the prediction of all or early FGR or PE+FGR, and this is consistent with the results of multiple regression analysis where both markers showed a high correlation with PE, but a small or no correlation with FGR or PE+FGR.

### 4.2. Interpretation of Results and Comparison with Findings of Previous Studies

Inhibin-A is a glycoprotein hormone that is abundantly expressed in the placenta, and its levels in both the placental and circulating maternal levels are increased in cases of PE, and the increase is apparent from the second trimester of pregnancy [23,26,27]. Inhibin-A was initially identified as a second trimester marker of chromosomal abnormalities [37] and was subsequently reported as a second and third trimester marker of PE [37–42]. We found

that the Inhibin-A level is considerably higher in early than late PE; the magnitude of increase was greater in PE alone rather than PE+FGR or FGR alone. Yet, our regression analysis showed no correlation of Inhibin-A level with gestational age or birth weight, which are classical parameters to define PE severity. These findings are consistent with previously reported results [38–42].

We have previously reported [23–25] that in PE and/or FGR, there is a reduction in the level of maternal serum PlGF and an increase in the level of sFlt-1 and of the sFlt-1/PlGF ratio. In the current study, we also examined the potential value of Inhibin-A/PlGF ratio, but this appeared to be less powerful than any of the other measures. In comparison, we showed that the combined effect of PlGF and inhibin-A or of sFlt-1/PlGF ratio plus Inhibin-A is superior to PlGF alone or to sFlt-1/PlGF ratio alone.

The PlGF and sFlt-1/PlGF results of our study are consistent with large-scale, high-quality studies by others [43–49]. Our secondary analysis focuses on the quantification of the added value of maternal serum Inhibin-A on top of the known value of maternal serum angiogenic markers. We showed that the main value of Inhibin-A is in augmenting the accuracy of predicting PE, which is a pregnancy complication that at least in our Slovenian cohort did not reach diagnostic accuracy by any combination of the angiogenesis markers on their own without adding Inhibin-A into the analysis. Neuman et al. [42] were the first to examine the added value of Inhibin-A to that of angiogenic markers and reported that this was beneficial mainly for early rather than late PE. In this study, we found that maternal serum Inhibin-A had an additive value to both PlGF and the sFlt-1/PlGF ratio in the prediction of both early and late PE and to a lesser extent of PE+FGR. We also checked the markers by multiple regressions, and the results indicated no correlations with gestational age or birth weight and marginal correlation with the MAP for sFlt-1/PlGF ratio. We clearly showed that the beneficial added value of Inhibin-A is two-fold; first, we found very high correlation for increased Inhibin-A and sflt-1/PlGF ratio for the diagnosis of PE and the diagnosis of PE+FGR, but not for FGR alone. Second, we found an increased correlation between the decreased PlGF and high concentration of Inhibin-A in the diagnosis of each of the three complications. Hence, we are expanding the conclusions of Neuman et al. [42].

In the past, cutoff values of PlGF and of sFlt-1/PlGF ratio were used to predict the short-term absence or presence of PE for clinical management of pregnancy-related complications [17–22,43–49]. In our study, Inhibin-A brought the accuracy to the diagnostic level in the complications of PE and for PE+FGR. The added value of Inhibin-A was clear for both the continuous and the cutoff models. For the cutoff model, the negative predictive values (NPVs) and the positive predictive value (PPV) reached above 93% in the case of pure PE. Thus, although the literature argues for the superiority of the continuous model [18,48], we concluded that although the continuous model might be a little more accurate, acting by cutoffs was very adequate especially for combining Inhibin-A with PlGF.

In the case of FGR, the accuracy level of PlGF alone and to a lower extent of the sFlt-1/PlGF ratio was exceptionally high to begin with, and hence added value by Inhibin-A was negligible. This is likely to be the consequence of our diagnostic criteria of FGR, which included the presence of small for gestational age fetuses with abnormal arterial and venous Doppler indices [50,51].

*4.3. Implications for Clinical Practice*

Inhibin-A has a large additive value to PlGF alone for increasing the accuracy of the diagnosis of PE, both for all cases and for those delivering < 34 weeks. Inhibin-1 is also very effective when added to the sFlt-1/PlGF ratio in predicting PE. As in our cohort, the diagnosis of PE by the angiogenesis markers or their ratio as stand-alone tools are less accurate ones. Hence, adding Inhibin-A appears advantageous for future clinical value in PE diagnosis. The added value for the diagnosis of all cases of FGR and PE+FGR is also meaningful, but for the early cases, the added value is marginal. If these findings are confirmed in larger studies, then the measurement of a pair made of PlGF with Inhibin-A may be an alternative to a pair of PlGF with sFlt-1.

The described immunodiagnostic methods can be completed within 60–90 min, and the assays are suitable for points of care both in maternity hospitals and community clinics. The cost of marker testing is approximately 50–60 Euro each. Hence, it makes no difference measuring a pair of PlGF and Inhibin-A or a pair of PlGF and sFlt-1. Measuring all three can be proportionally more expensive. However, we emphasize the beneficial use of Inhibin-A and PlGF combined, whereas the use of the Inhibin-A/PlGF ratio is inferior!

## 5. Limitations of the Study

The main limitations of the study are as follows. First, the biomarkers were not measured at fixed time points, but rather when the patients were admitted to the hospital or were seen in outpatient clinics; however, this reflects clinical reality. For the women who delivered at <34 weeks, there was no difference for the GA at enrollment, whereas biomarker differences were significant. Regarding gestational week, in all cases, enrollment was ≈3 weeks later for the term normal delivery group, but the GA was not different from PTD < 37. The biomarker level was significantly different between the complication groups to the term delivery control and the PTD < 37 weeks control, whereas the two control groups were not significantly different from one another. Hence, it appears that at this period of pregnancy, marker adjustment to GA would not marginally affect biomarker level. Future studies with a larger cohort will enable further verification of this point. A second limitation is that the design of the study was such that we did not perform repeated measurements during pregnancy, which were shown to improve the prediction accuracy.

## 6. Conclusions

Inhibin-A augments the accuracy of pro-and-anti-angiogenic markers in the prediction of suspected PE and PE+FGR around delivery. Further studies are warranted with larger cohorts of pregnant women to define the exact role of Inhibin-A in the prediction of these pregnancy complications.

**Author Contributions:** All authors were involved in writing and editing the manuscript. The study was conceived by J.O., K.K., T.P.S. and N.T., who wrote the study protocol and obtained ethics committee approval for the study. T.P.S., V.F.V. and N.T. enrolled the patients to the study, recorded their demographic characteristics and medical history, and performed the obstetric and sonographic evaluations. Analysis of samples was carried out by K.K., T.F. and J.O. The database was built and completed by K.K., A.S.-N. and H.M., who developed the mathematical models and performed the statistical analyses together with K.H.N., and J.O., K.H.N., H.M., J.O. and K.K. secured funding for this study and coordinated the cooperation among the study tems in Slovenia, the UK and Israel. All authors have read and agreed to the published version of the manuscript.

**Funding:** This study was supported in part by the ASPRE project (EU, FP7 # 601852, H.M., K.H.N.), and by the Graduate School of the University Medical Center, Ljubljana, Slovenia (K.K., J.O.).

**Institutional Review Board Statement:** The study was conducted according to the guidelines of the Declaration of Helsinki for Human subjects involved in clinical research, and was approved b, The National Medical Ethics Committee of the Republic of Slovenia involved in regulating human subjects participation in clinical trials approved the study on 4 December 2012 (Approval No. 104/04/12).

**Informed Consent Statement:** A written informed consent was obtained from each pregnant woman involved in the study.

**Data Availability Statement:** Data are available at the websites of Tel Hai College and the University Medical Center, Ljubljana, Slovenia. Due to cyber security issues, links to the websites should be pre-arranged in advanced from A.S.-N. (Tel Hai College) and from K.K. or J.O. (University Medical Center, Ljubljana).

**Conflicts of Interest:** The authors report no conflict of interest. The authors alone are responsible for the content and writing of the paper. The funders had no role in the design of the study; in the collection, analyses, or interpretation of the data; in the writing of the manuscript, or in the decision to publish the results.

## Abbreviations

| | |
|---|---|
| ACOG | American College of Obstetrics and Gynecology |
| AUC | Area under the curve of the receiver operation characteristic curve |
| BMI | Body mass Index |
| BP | Blood pressure |
| dBP | Diastolic blood pressure |
| DR | Detection rate (sensitivity) |
| FMF | Fetal Medicine Foundation |
| FGR | Fetal growth restriction |
| FPR | False positive rate (1-specificity) |
| ISSHP | International Society for the Study of Hypertension disorders of Pregnancy |
| ISUOG | International Society of ultrasound in Obstetrics and Gynecology |
| IVF | In vitro fertilization |
| MAP | Mean arterial blood pressure |
| NPV | Negative predictive value |
| PE | Preeclampsia |
| PlGF | Placenta growth factor |
| PPV | Positive predictive value |
| PTD | Preterm delivery |
| ROC | Receiver operation characteristic curve |
| UTPI | Uterine Artery Pulsatility Index |
| sBP | Systolic blood pressure |
| sFlt-1 | Soluble Fms-like tyrosine kinase 1 |
| 95% CI | 95% Confidence Interval |

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
