# Peer review of "Maternal Serum Inhibin-A Augments the Value of Maternal Serum PlGF and of sFlt-1/PlGF Ratio in the Prediction of Preeclampsia and/or FGR Near Delivery—A Secondary Analysis"

_2673-3897, doi:10.3390/reprodmed2010005_

Round 1
Reviewer 1 Report
The manuscript of Adi et al. describes the usefulness of adding an immunoassay measurement of Inhibin-A from the maternal serum in addition to well-studied and widely-used angiogenic biomarkers, namly sFlt1 and PlGF, in light of predicting preeclampsia (PE) and/or fetal growth restriction (FGR). The authors showed a small improvement of predicting complicated pregnancies, especially PE, when combined with sFlt1/PlGF ratio, which is inline with the previous study reported by others. However their study design and description of sample were not clearly presented. Especially this reviewer found it is very worrying that the gestational ages (GA) when the markers were measured/assayed were not presented. It may be that the authors did not design their study carefully enough to take GA of the assay into account when the comparisons were made between cases and controls. Also the manuscript needs more improvements especially with regard to presenting their results. That said, the manuscript does not warrant a publication as its current form, but it could be potentially beneficial to reach wider readers who share the interests.
Major points:
- The number of healthy control group is relatively low compared to affected cases - how were they selected from the study cohort? Having said that, the description of the study design is not very clear with regard to the selection and exclusion criteria. For example, what is the total size of cohort and how many of them were studied in the current report?
- It is also not clear when (i.e in which GA) the maternal blood samples were collected. For example, when the comparisons were made (e.g. Figure 1 and Table 1), 1) how variable is the GA (when the markers were measured) within and across the groups, and 2) could the results be reproduced when the comparisons were made by adjusting GA of the assay. In ‘Limitations of the study’ section (line 267-272), the authors acknowledged that the measurements were not fixed at a given time point of gestation. However the GA could be presented and taken into account when the measurements of markers were compared.
- In Figure1:
- Are the cases from the ‘all patients’ group or those from the ‘Birth < 34 weeks’ group? It is recommended to have the sample size in the figure legend or in figure itself.
- It is very odd that the measurement of PlGF is presented as µg/mL, whereas it is shown as pg/mL for Inhibin A. Given the ratio of Inhinin A / PlGF in Table 1, the unit of PlGF must be pg/mL. This is also reflected as such in the cuff-off values of Table 2, which is given as ‘pg/mL’.
- In Table 1:
- The numbers of samples shown in Table 1 do not match with those described in line 39 - authors should mention and explain the discrepancy.
- There are unannotated marks (‘a’ and ’b’) from the PlGF row of ‘all participants’.
- The square brackets should be closed for “3.1 [0.7-3.6[“ and “1.1 [0.2-3.5[“.
- No units given to PlGF and Inhibin A.
- Most importantly, the “sFlt-1 / PlGF” values of <34 weeks birth cohort are too low to be true for PE and FGR cases. They are even lower than that of ‘Birth <37 weeks’, so the p-values cannot be true for such marginal difference between the adverse outcome groups and ‘birth <37 weeks’. It is likely that “sFlt-1 / PlGF” of cases could be wrong.
- Having observed a lower level of Inhibin A in unaffected control group, could the authors possibly comment why it appears lower in ‘Birth <37 weeks’ than ‘Term delivery’.
- Also Table 1 should be reformatted to make it more readable. For example, 1) the number of samples could be presented in a separate row from the first row, 2) ‘birth < 34’ could be decoupled with the numbers of samples and as shown from the ‘All participants’ row In addition, other maternal and fetal characteristics could be added (please refer to the table 1 from Sovio et al. 2017 (PMID: 28167687)).
- Line 143-145, the cut-off values of ‘good’ separation should be justified.
- Line 118, “Cut-offs were marked as X on the ROC curves” - no marks found in Figure 2 and Figure 3.
- In Table 3:
- This table could be merged into Table 2 as some pieces of information are redundant what are shown in Table 1 (e.g. PlGF and sFlt1/PlGF)
- Make sure that the bracket is closed for AUC range.
- What does the column ‘p’ stand for? If it represents p-value, what exactly does it mean (e.g. what is it compared to)?
- “Multiple regression” (line 175): please describe how the models (i.e. equations) were generated. Also, describe “B”, “S.E.”, “β”, and “F(6,63)” in the legend of Table 4.
- Implications for clinical practice (lines 260-266): it would be great to discuss the cost of adding Inhibin A.
- Maining findings (lines 200-219): descriptions of adding Inhibin A could be more focused on its comparison to sFlt1/PlGF, which is more clinically useful rather than PlGF alone.
- Lines 263-264, “the measurement of Inhibin A may be alternative to that of sFlt-1”: this seems to imply that Inhibin-A/PlGF could be an alternative to sFlt-1/PlGF. It could be potentially misleading as Inhibin-A/PlGF has a very marginal effect compared to sFlt-1/PlGF.
Minor points:
- Line 215, ‘seventh’ should be replaced with ‘sixth’.
- Line 267-272, “Limitations of study” section is preferred to be within Discussion
- Line 67, FMF-like => Fms-like
- Ref #25 => submitted
- Lines 71074 do not read well.
- Line 305 FGR - Fetal Growth Restriction
- Line 102, PLGF => PlGF
- Some unused abbreviations should be removed, for example, ALT, AST, FMF (except line 67 where it should be replaced with Fms), IVF, and LDH.
- Line 112, Mann-Whiney test also performed by SPSS?
Author Response
Review Report Form
Reviewer 1
Comments and Suggestions for Authors
The manuscript of Adi et al. describes the usefulness of adding an immunoassay measurement of Inhibin-A from the maternal serum in addition to well-studied and widely used angiogenic biomarkers, namely sFlt1 and PlGF, in light of predicting preeclampsia (PE) and/or fetal growth restriction (FGR). The authors showed a small improvement of predicting complicated pregnancies, especially PE, when combined with sFlt1/PlGF ratio, which is inline with the previous study reported by others.
Answer: We thank the referee.
However, their study design and description of sample were not clearly presented. Especially this reviewer found it is very worrying that the gestational ages (GA) when the markers were measured/assayed were not presented. It may be that the authors did not design their study carefully enough to take GA of the assay into account when the comparisons were made between cases and controls.
Answer: We thank the referee for highlighting this issue. The gestational age (GA) is now entered into Table 1 as requested by the referee later in his comments. We enrolled cases at the time they attended with the suspected complications. To account for the range of GAs, we also enrolled cases of preterm delivery (PTD) < 37 wks who had none of the study complications (preeclampsia (PE), fetal growth restriction (FGR) or PE+FGR). The included patients had no fetal abnormalities and no chorioamnionitis. As it depicted in Table 1, the biomarker levels of PTD <37 wks control group are not statistically different from the unaffected term delivery control group, but the marker level of each of the complication group was statistically different from their level in both control groups (Table 1). GA comparison in Table 1 shows that for the early cases that delivered at <34 wks there was no difference in the GAs at enrollment (or at delivery). When all cases of each complication group were compared to term delivery control group, the GA of the latter was on average 3 weeks earlier. The GA of the control group of PTD <37 was not different from the GA of the complication groups (Table 1). Future studies with a larger cohort will enable further verification of the contribution of marker level adjustment to GA. This was impossible in this cohort as such adjustment requires at least 30 patients for each GA (Cuckle Placenta. 2011 PMID: 21257082.) We addressed this issue in the study limitation section (lines 318-326).
Also, the manuscript needs more improvements especially with regard to presenting their results. Answer: We hope the revised manuscript includes improved data presentation
That said, the manuscript does not warrant a publication as its current form, but it could be potentially beneficial to reach wider readers who share the interests.
Answer: We hope the revised manuscript does meet the interest of readers about the usefulness of biomarkers in the management of preeclampsia, which is the focus of this journal issue.
Major points:
The number of healthy control group is relatively low compared to affected cases - how were they selected from the study cohort? Having said that, the description of the study design is not very clear with regards to the selection and exclusion criteria. For example, what is the total size of cohort and how many of them were studied in the current report?
Answer: The issues were restated in the revised manuscript
Term delivery controls were recruited in the same high risk pregnancy delivery clinics as the complication groups, but they didn’t developed PE and FGR or both FGR+PE. The exclusion and inclusion criteria are better defined (Lines 91-94) along with further clarification on the features of the complication groups (lines103-111). Cohort size was 125 patients. All patients attending the clinic who met the inclusion and exclusion criteria and were willing to sign the informed consent were enrolled. As our population was small, we could extract all the data for each so there were no loss to follow up and none drop consent. (lines141-144)
It is also not clear when (i.e in which GA) the maternal blood samples were collected. For example, when the comparisons were made (e.g. Figure 1 and Table 1),
Response: All patients were tested at the time of enrolment and the comparison was made accordingly (line 146-148). This is now also clarified in Table 1 and in the legends to Figure 1.
- how variable is the GA (when the markers were measured) within and across the groups
Answer: We entered the GA into Table 1 to clarify the issue, and also see answer above,
2) could the results be reproduced when the comparisons were made by adjusting GA of the assay.
Answer: The issue is discussed in details at the beginning of the response to the comments of the referee. Marker level was not adjusted to GA because of cohort size limitation given that the standard adjustment requires at least 30 patients for each GA (Cuckle Placenta. 2011 PMID: 21257082.). The issue will be further examined in subsequent larger cohort.
The figure of marker level versus GA for PLGF is shown below. It indicates a clear large difference between the biomarker level distribution of control (made from term and preterm delivery) and complication cases
In ‘Limitations of the study’ section (line 267-272), the authors acknowledged that the measurements were not fixed at a given time point of gestation. However, the GA could be presented and taken into account when the measurements of markers were compared.
Answer: The referee is right. In the revised version we entered the gestational age into Table 1, and further elaborated on this issue in the limitation section (Lines 318-326).
In Figure1:
Are the cases from the ‘all patients’ group or those from the ‘Birth < 34 weeks’ group? It is recommended to have the sample size in the figure legend or in figure itself.
Answers: The figure represents all cases of the pathology that also include the ones delivered <34 wks. Table 2 provides details for all cases versus the subgroup who delivered < 34 wks. Sample size was added to the table and the figure legend. (Table 2, Figure 1)
It is very odd that the measurement of PlGF is presented as µg/mL, whereas it is shown as pg/mL for Inhibin A. Given the ratio of Inhibin A / PlGF in Table 1, the unit of PlGF must be pg/mL. This is also reflected as such in the cuff-off values of Table 2, which is given as ‘pg/mL’.
Answers: Sorry. The referee is right. All markers were calculated in pg/ml. Units have now been corrected. (Figure 1).
In Table 1:
The numbers of samples shown in Table 1 do not match with those described in line 39 - authors should mention and explain the discrepancy.
Answer. Numbers were fixed (Abstract Lines 31-33, Results Lines 141-144, Table 1).
There are unannotated marks (‘a’ and ’b’) from the PlGF row of ‘all participants’.
Answer: The unannotated marks were removed. (Table 1)
The square brackets should be closed for “3.1 [0.7-3.6[“ and “1.1 [0.2-3.5[“.
Answer: Brackets were corrected across the manuscript as indicated
No units given to PlGF and Inhibin A.
Answer: Units were added to Table 1 also line 163
Most importantly, the “sFlt-1 / PlGF” values of <34 weeks birth cohort are too low to be true for PE and FGR cases. They are even lower than that of ‘Birth <37 weeks’, so the p-values cannot be true for such marginal difference between the adverse outcome groups and ‘birth <37 weeks’. It is likely that “sFlt-1 / PlGF” of cases could be wrong.
Answer: Values were corrected (Table 1)
Having observed a lower level of Inhibin A in unaffected control group, could the authors possibly comment why it appears lower in ‘Birth <37 weeks’ than ‘Term delivery’.
Answer: Inhibin A level in term delivery is 724 [491-904] pg/ml (n=21), and in preterm delivery < 37wks it is 330 [261-928] (n=12). The two are not significantly different, maybe due to the sample size. In comparison, the level in PE is 2,097 [1,546-2,660] and these results are significantly higher compared to any of the former ones (p<0.05)* (Table 1)
Also, Table 1 should be reformatted to make it more readable. For example,
- the number of samples could be presented in a separate row from the first row,
Answer: A separate line was added for the number of patients for both the entire complication group and to the sub-groups <34 weeks (Table 1).
- ‘birth < 34’ could be decoupled with the numbers of samples and as shown from the ‘All participants’ row. In addition, other maternal and fetal characteristics could be added (please refer to the table 1 from Sovio et al. 2017 (PMID: 28167687)).
Answer: The entire Table 1 was modified more or less according to Sovio et al. 2017 (PMID: 28167687). Our cohort was way smaller, and the medical and lab team was able to complete all patients’ information so there was no missing data. We organized the table with the first set of entries relevant to prior risk, then marker levels and then delivery outcome. The number of patients in each group was added. All patients were Caucasians; hence, ethnicity was not included in Table 1 but added to the sample description (line 96). The table is divided to all patients on top and the subgroups who delivered before 34 wks in the bottom. There was not much point to add a description of the patients’ division between quartiles, given the relatively small patients number in each group. Units of biochemical markers were added (pg/ml). (Table 1, Table 2)
Line 143-145, the cut-off values of ‘good’ separation should be justified.
Answer: The good separation by the cutoffs places >90% of the unaffected and of the PTD group (corresponding to 10% FPR) in one side of the cutoff line and a large majority of the patients with complications were in the other side of the line. (lines 164-169)
Line 118, “Cut-offs were marked as X on the ROC curves” - no marks found in Figure 2 and Figure 3.
Answer. The referee is right. The X mark was included in earlier draft, and we forgot to omit it from the methods. Now it is omitted.
In Table 3:
This table could be merged into Table 2 as some pieces of information are redundant what are shown in Table 2 (e.g. PlGF and sFlt1/PlGF). Make sure the brackets is closed for AUC range.
Answer: The repeated values are placed to ease on the reader to see the differences between a single and combined analysis. The merging of the two tables appears really unpleasant to the reader.
Brackets were closed for all ROCs.
What does the column ‘p’ stand for? If it represents p-value, what exactly does it mean (e.g. what is it compared to)?
Answer: In Table 3 the column P represent the significance compared to the arbitrary midline curve of AUC=0.5. This is now explained in Table 3 legend and in Figure 3.
“Multiple regression” (line 175): please describe how the models (i.e. equations) were generated. Also, describe “B”, “S.E.”, “β”, and “F(6,63)” in the legend of Table 4.
Answers: The definition of “B”, “S.E.”, “β”, and “F(6,63)” were added into the legend of Table 4 and to the text lines 185-190.
The equations were generated by the SPSS software according to the values entered.
Implications for clinical practice (lines 260-266): it would be great to discuss the cost of adding Inhibin A.
Answers: Cost implications of adding inhibin A were added to the discussion (lines 311-314)
Main findings (lines 200-219): descriptions of adding Inhibin A could be more focused on its comparison to sFlt1/PlGF, which is more clinically useful rather than PlGF alone.
Answers: The potential clinical value of using two markers (inhibin A and PlGF) is compared to the case of PlGF alone and to the case of the pair of sFlt-1/PlGF ratio (lines 273-287). Also, we discussed the use of three markers (Inhibin and sFlt-1/PlGF +inhibin) compared to two sFlt-1/PlGF. The accuracy is discussed in lines 275-287 as mentioned above and the cost in lines 311-314)
Lines 263-264, “the measurement of Inhibin A may be alternative to that of sFlt-1”: this seems to imply that Inhibin-A/PlGF could be an alternative to sFlt-1/PlGF. It could be potentially misleading as Inhibin-A/PlGF has a very marginal effect compared to sFlt-1/PlGF.
Answers: We are sorry to confuse the reviewer. Actually, we found Inhibin A/PlGF ratio inferior to sFlt-1/PlGF ratio (line 273-275). In comparison, combining the performance of inhibin-A and PlGF (not their ratio) or Inhibin-A and sFLT-1/PlGF appears to be superior to PlGF alone or sFlt-1/PlGF in the case of PE. The matter is rewritten to clarify (line 274-287).
Minor points:
Line 215, ‘seventh’ should be replaced with ‘sixth’.
Answer: Corrected (line 255)
Line 267-272, “Limitations of study” section is preferred to be within Discussion
Answer: Sorry, the formation is according to the journal guidelines.
Line 67, FMF-like => Fms-like
Answers: Corrected (line, 25,73,202,206,212,218)
Ref #23 => submitted
Answers: The referee wrote 23 but 23 and 24 were actually published. 25 was submitted to publications and this is now defined (line75,100,273)
Lines 71-74 do not read well.
Answers: Corrected (line 77-79)
Line 305 FGR - Fetal Growth Restriction
Answers: Corrected (line 357)
Line 102, PLGF => PlGF
Answer: Corrected (line 115)
Some unused abbreviations should be removed, for example, ALT, AST, FMF (except line 67 where it should be replaced with Fms), IVF, and LDH.
Answer: We removed ALT AST and LDH that were not used here (standard laboratory blood test of enzymes associated with HELLP syndrome. FMF – Fetal Medicine Foundation remained in as we used it. Fms was added. IVF remained in as it was entered to Table 1.(Lines 350-370)
Line 112, Mann-Whiney test also performed by SPSS?
Answer: Yes, The SPSS generates Mann-Whiney test results (line 127)
Reviewer 2 Report
This is a very interesting study. However, it would be important for the authors to address some questions
INTRODUCTION
Lines 53-54: please rephrased “Preeclampsia refers to the new onset of hypertension and proteinuria and usually begins after 20 weeks of pregnancy in women whose blood pressure had been normal.
Lines 58-59: please be more accurately. Fetal growth restriction (FGR) describes the fetus that does not grow to its expected biological potential in utero, and is a relatively common complication of pregnancy. FGR, is a pathological condition wherein the placental fails to deliver an adequate supply of oxygen and nutrients to the developing fetus, termed placental insufficiency. The individual's genetic makeup, nutrient availability from the mother, and environmental factors, coupled with the capacity of the placenta to adequately transfer nutrients and oxygen to the fetus, and endocrine modulation of these interactions are the basic determinants of fetal growth.
Method
The authors did not report when the biomarkers were measured. As they report this fact could be a serious limitation for the study.
Author Response
Reviewer 2.
Comments and Suggestions for Authors
This is a very interesting study. However, it would be important for the authors to address some questions
Response: Thank you
INTRODUCTION
Lines 53-54: please rephrase “Preeclampsia refers to the new onset of hypertension and proteinuria and usually begins after 20 weeks of pregnancy in women whose blood pressure had been normal.
Response: Modified as suggested (lines 53-55)
Lines 58-59: please be more accurate. Fetal growth restriction (FGR) describes the fetus that does not grow to its expected biological potential in utero, and is a relatively common complication of pregnancy. FGR, is a pathological condition wherein the placental fails to deliver an adequate supply of oxygen and nutrients to the developing fetus, termed placental insufficiency. The individual's genetic makeup, nutrient availability from the mother, and environmental factors, coupled with the capacity of the placenta to adequately transfer nutrients and oxygen to the fetus, and endocrine modulation of these interactions are the basic determinants of fetal growth.
Response: Modified as suggested (lines 58-65)
METHOD
The authors did not report when the biomarkers were measured. As they report this fact could be a serious limitation for the study.
Response: Thank you, now added.(Lines 114-115, Table 1)
Reviewer 3 Report
Dear Authors,
Your topic is actual, bur unfortunately the same conception has been observed by Prof. A. Grafka and published ("PLFG and sFLT-1 in clinical diagnosis of preeclampsia") 7 years ago. I can recommend you to read this article and add one more citation.
Introduction. It is not sufficient, more explanation of a preeclampsia and fetal growth restriction should be added. Not all abbrevations have been described previously. Please, give a short explanation to sFLT-1.
You wrote about pre-term delivery. It is easy for obstetricians, but should be explained for others, because of heterogenity of the audience.
Samlpe and methods. I have a question concerning a medical history. Are your patients have some chronic diseases?
Discussion. PIGF and sFlt-1 ration vs other markers (not only Inhibin A) - do you have such information or maybe results of your previous studies.
Thank you.
Author Response
Third Reviewers
Comments and Suggestions for Authors
Dear Authors,
Your topic is actual, bur unfortunately the same conception has been observed by Prof. A. Grafka and published ("PLFG and sFLT-1 in clinical diagnosis of preeclampsia") 7 years ago. I can recommend you to read this article and add one more citation.
Answer: We are extremely sorry. We searched pubmed many times and also run a few and googles. We could not find any paper by Grafka A (“PLFG and sFLT-1 in clinical diagnosis of preeclampsia") from 7 years ago - neither from in 2013, 2014 or 2015. Very sorry indeed. If in a second review the referee will provide more information, we will be VERY VERY happy to add the reference. We just want to say, judging from the reference title, that our main topic is not PlGF or sFlt-1 or their ratio, which was a subject to a different manuscript (reference 25) submitted earlier to this journal, but we focused on the added value of Inhibin A to the accuracy of prediction by PlGF or by sFlt-1/PlGF ratio.
Introduction. It is not sufficient, more explanation of a preeclampsia and fetal growth restriction should be added.
Answer: The description of preeclampsia and FGR is rewritten (lines 53-65)
Not all abbreviations have been described previously
Answer: All abbreviation are now listed at the end of the manuscript .(line 350-372) and each time they appear in the tables and figures
Please, give a short explanation to sFLT-1.
Answer: Explanation was added (lines 71-76).
You wrote about pre-term delivery. It is easy for obstetricians, but should be explained for others, because of heterogenity of the audience.
Answer: The referee is right. A more detailed explanation for the type of preterm delivery cases included is now added. (lines 110-114).
Sample and methods. I have a question concerning a medical history. Are your patients have some chronic diseases?
Answer: In the revised manuscript, the medical and pregnancy history were added to Table 1.
Discussion. PIGF and sFlt-1 ration vs other markers (not only Inhibin A) - do you have such information or maybe results of your previous studies.
Answer: In a previous published study, reference 23, Sharabi-Nov A, Kumar K, Fabjan Vodušek V, Premru Sršen T, Tul N, Fabjan T, Meiri H, Nicolaides KH, Osredkar J. Establishing a differential marker profile for pregnancy complications near delivery. Fetal Diagn Ther 2020; 47: 471-484. We provided a description of nine biomarkers (PlGF. sFlt-1. sFlt-1/PlGF ratio, sEndoglin, PlGF/sEndohlin+sFLT-1), TNF alpha, VEGF, Inhibin A, and PP13), and of several biophysical markers (uterine artery pulsatility index and resistance index measured by Doppler, mean arterial blood pressure, and two markers of endothelial stifness measured by EndoPAT). In this paper we focused on inhibin A and on two additional markers, that in either the one cited and the one in review by this journal were found the best ones, and evaluated if inhibin could provided further accuracy.
Submission Date 31 December 2020
Date of this review 20 Jan 2021 11:44:52
Round 2
Reviewer 1 Report
The authors addressed concerns and recommendations raised by this reviewer. The manuscript appeared improved than the previous version. I would like to appreciate their work and congratulate to the authors.
Author Response
Reviewer 1:
The authors addressed concerns and recommendations raised by this reviewer. The manuscript appeared improved than the previous version. I would like to appreciate their work and congratulate to the authors.
Answer; Thank you
